# The Human Ovary and Future of Fertility Assessment in the Post-Genome Era

**DOI:** 10.3390/ijms20174209

**Published:** 2019-08-28

**Authors:** Emna Ouni, Didier Vertommen, Christiani A. Amorim

**Affiliations:** 1Pôle de Recherche en Gynécologie, Institut de Recherche Expérimentale et Clinique, Université Catholique de Louvain, 1200 Brussels, Belgium; 2PHOS Unit, Institut de Duve, Université Catholique de Louvain, 1200 Brussels, Belgium

**Keywords:** mass spectrometry, ovary, fertility, biomarkers, oocyte competence

## Abstract

Proteomics has opened up new avenues in the field of gynecology in the post-genome era, making it possible to meet patient needs more effectively and improve their care. This mini-review aims to reveal the scope of proteomic applications through an overview of the technique and its applications in assisted procreation. Some of the latest technologies in this field are described in order to better understand the perspectives of its clinical applications. Proteomics seems destined for a promising future in gynecology, more particularly in relation to the ovary. Nevertheless, we know that reproductive biology proteomics is still in its infancy and major technical and ethical challenges must first be overcome.

## 1. Introduction

Proteomics is an emerging discipline that involves studying the proteome, namely the gene expression of a cell, tissue or organism, through analyzing proteins and their subsequent translational modifications by mass spectrometry (MS). The proteomic bottom-up strategy (proteolytic peptide mixture analysis) is the most commonly used method for analysis of biological samples. Strategies applied to prepare proteins or more complex proteomic samples for MS analysis involve many steps, and in bottom-up proteomics, the protein constituent is first scaled down into peptides, either by chemical or enzymatic digestion, prior to MS analysis. Peptides are then separated by liquid mono- or two-dimensional chromatography to fractionate samples and thereby reduce their complexity and dynamic range. One of the parameters that comes into play in the quality of obtained results relates to the wide range of protein concentrations in studied samples. Indeed, proteins present in low copy numbers in samples would be hard to identify without a prior step of enrichment [1].

MS analysis is then performed on individual peptides, and data are merged and consolidated to reveal the protein identity and/or its characteristics [1]. This automated process is greatly facilitated by the use of well-annotated proteome databases and bioinformatics (Figure 1).

In the post-genome era, while analysis of mRNA expression remains as the technique of choice to elucidate mechanisms of function and regulation in ovarian tissue, the use of proteomics is still relatively limited. Although genomics provides valuable information on certain biological functions, the proteome is the complete representation of proteins expressed by a genome. It is therefore more representative of the phenotype and able to provide complementary information that is yielded by gene expression studies. Indeed, gene expression studies of RNA transcripts often cannot predict the abundance or function of proteins or their post-translational modifications [3].

Unlike other techniques, proteomics can identify hundreds or even thousands of proteins in the same experiment from the same sample, giving access to a whole range of potentially interesting proteins and allowing them to be quantified relatively [4]. Thus, it allows investigation without a priori knowledge, in contrast to targeted studies using specific antibodies.

In the last decade, growing interest in proteomic approaches in gynecology [5] has aimed to (i) define biomarker profiles for appraisal of oocyte quality to improve success rates in in vitro fertilization (IVF), (ii) limit the complications of high-risk pregnancies, (iii) create proteomic maps for biomarker identification and (iv) fine-tune ovarian tissue-engineered models.

## 2. Proteomics for Selection of Competent Embryos and Oocytes in IVF

Since the birth of the first IVF baby, assisted procreation techniques have come a long way. Despite these advances, it has been reported that 70–80% of in vitro-produced embryos fail to implant, and 66% of IVF cycles do not result in pregnancy [6].

### 2.1. Follicular Fluid: An Inexhaustible Biosource for Investigation of Competence Biomarkers

Oocyte and embryo selection are principally based on morphological criteria, despite the contentiousness of morphological evaluation and its limitations [7]. Since human ovarian follicular fluid constitutes the in vivo microenvironment for oocytes during folliculogenesis and is also one of the most abundant ‘waste products’ in assisted reproduction, it is a readily available source of biomolecules for competence evaluation of collected oocytes and embryos. This noninvasive technique could be used to predict IVF success rates [7]. Indeed, understanding the protein expression profile of follicular fluid and the functional correlations between its components and corresponding IVF results could play a key role in the development of new methodologies, which would not only target oocyte and embryo viability, but also predict the chances of success in assisted reproduction programs.

Many research teams have focused their efforts on the proteomic profiling of follicular fluid and, through proteomic and bioinformatic analysis, have found that the main signaling pathways involving identified proteins are chiefly related to inflammation, coagulation, tissue remodeling and lipid metabolism [8,9]. One of the advantages of using proteomics is being able to correlate the detected proteins with their corresponding signaling pathways to better understand their role (Figure 1).

#### 2.1.1. Role of Inflammation and Coagulation Cascades in IVF

Ovulation is an inflammation-like process during which the ovarian balance is achieved through pro-inflammatory action mediated by cytokines (such as interleukins IL-1, IL-6 and IL-8, and tumor necrosis factor alpha (TNF-α)) and negative feedback from anti-inflammatory cytokines, accompanied by leukocyte recruitment by monocyte chemotactic protein-1 (MCP-1) [9].

Balanced and chronologically regulated cytokine activity is an absolute requisite for physiological maturation and follicle rupture. By contrast, aberrant cytokine activity could significantly affect follicle dynamics, as is the case with TNF-α. High intrafollicular levels of TNF-α have been linked to insufficient development of oocytes and inhibition of their maturation, which has led to a decline in their quality [10]. Inappropriate complement activation has been correlated with an increased risk of miscarriage. Nevertheless, the functions of the complement and coagulation cascade during IVF are still being debated. While some studies show that inhibition of complement activation improves angiogenesis and saves pregnancies [8], others, by comparing human follicular fluid from fertilized and unfertilized oocytes, reveal higher concentrations of C3 in fertilized oocytes [11]. This contradiction may be partially explained by the involvement of C3, C4 and C9, as well as complement factor H and clusterin, in complement inhibition in women following controlled ovarian stimulation for IVF [11]. However, the role of coagulation and inflammation cascades during IVF requires further investigation.

Proteins of the complement system promote coagulation through inhibition of fibrinolysis; in return, coagulation enhances complement activation. Key players in this synergy are plasmin and thrombin. During follicular development, proteins involved in coagulation in follicular fluid contribute to its liquefaction, fibrinolysis and rupture of the follicle wall. Deregulation of fibrinolysis has been correlated with implantation failure and recurrent abortions [9].

#### 2.1.2. Impact of Extracellular Matrix Turnover on Oocyte Quality

Numerous studies have demonstrated the involvement of coagulation proteins in tissue remodeling and repair, namely the effect of thrombin and plasmin on the activity of other serine proteases and matrix metalloproteinases (MMPs) responsible for extracellular matrix (ECM) turnover [7,9]. Proteomic analysis of follicular fluid coupled with bioinformatics has successfully revealed potential major players in the evaluation of oocyte quality. These proteins of interest interact indirectly with thrombin and plasmin and are implicated in modifications to the pre- and postovulatory ECM. For example, plasmatic carboxypeptidase B2 (CPB2) and hyaluronan-binding protein 2 (HABP2) are involved in tissue repair and coagulation in response to ovulation [7,8]. While CPB2 controls lysis of blood clots, HABP2 activates extrinsic coagulation pathways and the fibrinolysis-inducing urokinase enzyme [8].

Among the serine proteases potentially of interest for predicting oocyte quality is kallikrein kininogenase (KKS) [12]. This family of proteins possibly plays a role in the formation of the thecal vascular network, the contraction of the follicle walls during ovulation and the vascularization of avascular follicles after release of the oocyte [7]. Moreover, some serine protease inhibitors detected in the latest proteomic studies have emerged as indicators of oocyte quality, such as alpha-1-antitrypsin (A1AT) in mature follicles. Indeed, the inflammatory process that accompanies ovulation and is associated with release of reactive oxygen species (ROS) induces polymerization of A1AT and deactivates it accordingly, which is essential for oocyte release. We may then conclude that A1AT plays a primary role in the follicle maturation and temporal control of mature oocyte release [7].

Other effector proteins implicated in ECM degradation and remodeling and involved in follicle growth, angiogenesis, apoptosis and embryo implantation are MMPs, namely MMP-9, MMP-12, MMP-13 and MMP-25 [9]. The link between regulation of MMP activity and ovarian activity emphasizes the importance of ECM homeostasis to fertility, follicle growth and ovulation control. Indeed, MMPs control the bioavailability of hormones, growth factors and osmotically active molecules, as well as cellular response [7,9]. Experiments have shown that unregulated MMPs may have adverse effects on oocyte development and maturation, causing excessive degradation not only of the follicular ECM and resident growth factors and co-factors, but also of surface receptors in follicular somatic cells, and potentially the maturing oocyte itself [13].

Aberrant MMP activity could also have a significant impact on the cumulus–oocyte complex and the integrity of the zona pellucida. Because of the metabolic dependence of the growing oocyte on cumulus cells and the firm control exerted by the oocyte on the physiology of these cells, the functional integrity of tight junctions (gap junctions) is fundamental to the development and maturation of oocytes. In this case, uncontrollable MMP activity could degrade intra-cumulus and cumulus–oocyte junctions, damaging the maturation of oocytes and hence their required fertilization competence. MMP access to the zona pellucida may also lead to partial degradation, affecting interaction and fertilization of spermatozoa [7,9,13]. Finally, disruptions to the MMP/TIMP (tissue inhibitors of MMPs) ratio have been reported to be negative predictors of oocyte release, fertility and IVF success [9,14,15].

#### 2.1.3. Involvement of Lipid Metabolism in the Success of IVF

Another major class of proteins identified by follicular fluid proteomic strategies are those active in lipid metabolism, including the vitamin D receptor (VDR) and the retinoid X receptor alpha (RXR-α) heterodimer. These receptors belong to the family of steroid/thyroid hormone nuclear receptors, and their heterodimerization regulates expression of genes encoding proteins involved in various cellular functions: proliferation, differentiation, apoptosis or even angiogenesis. Curiously, VDR/RXR-α controls the MMP/TIMP balance, plasminolysis, fibrinolysis and other proteolytic activities by different mechanisms [9].

Clearly, qualitative and/or quantitative variations in these proteins between follicular fluid samples could affect follicle and oocyte development and maturation during IVF. Their use as biomarkers to assess the quality of oocytes is therefore justified. In addition, some follicular fluid proteins may change quantitatively because of different patient responses to hyperstimulation protocols and different causes of infertility. Use of proteomics in this context could therefore have a significant impact, not only on determining the quality of oocytes, but also on the development of personalized fertilization treatment programs.

### 2.2. A New Way of Predicting IVF Success: The Cumulus–Oocyte Complex

In addition to follicular fluid analysis by MS, the team of Braga et al. (2016) was interested in investigating the cumulus-oocyte complex [6]. Among proteins exclusively identified both in blastocysts and oocyte cumulus cells that have generated successful pregnancies, they found glutathione S-transferase A4 (GSTA4), heavy chain Ig V-III region TUR, protein disulfide-isomerase (PDI) and proto-oncogene serine/threonine-protein kinase mos. This study also demonstrated that overexpression of GSTA4 is implicated in protecting cells against apoptosis, while overexpression of glutathione S-transferase P (GSTP) may be associated with the c-Jun N-terminal kinase signaling pathway (JNK), shielding cells from death signals or oxidative stress [6]. Expression of certain proteins from the glutathione S-transferase (GST) family is promoted through mitogen-activated protein kinase (MAPK) signaling pathways as a self-defense response against toxins and growth factors. It is essential to stress the involvement of p38 MAPK at the level of reproductive cells in oocyte maturation and steroidogenesis. The results of this study suggest that proteins identified in cumulus cells may play a role not only in folliculogenesis, but also in the immune response. The importance of the immune system to embryonic development and reproduction remains unproven, but it appears to serve a function in the implantation of embryos.

Another protein whose key role was highlighted in this study is the proto-oncogene serine/threonine-protein kinase mos. In addition to being vital for the initiation of oocyte maturation, progression of meiosis I and II and arrest of meiosis in metaphase, it is also potentially important for assessing oocyte quality, embryonic development and implantation.

Since communication between the oocyte and cumulus cells is essential for acquisition of oocyte competence, proteins identified in the course of proteomic characterization of these cells could have therapeutic targets and may be used for enrichment of culture media. This characterization also provides a clearer picture of the physiology of in vivo maturation during controlled stimulation cycles. Furthermore, a proteomic study of cumulus cells might well be useful for predicting successful pregnancies and identifying patients who should be included in extended embryo culture programs or those who could benefit from cleavage-stage embryo transfer [6].

Despite progress in IVF at the clinical level, embryo selection is still based on unsatisfactory morphological criteria. A more appropriate selection of proteomic-based oocytes and competent embryos could enhance IVF success rates. However, while a number of biomarkers have been identified in basic research, these data are not used clinically. It is therefore essential to translate these new findings into clinical settings that could offer patients a significantly better chance of conceiving through IVF.

## 3. Ensuring Normal Pregnancy in Patients with Polycystic Ovary Syndrome through Proteomics

Polycystic ovary syndrome (PCOS) is often a source of pregnancy complications in patients with ovarian hyperstimulation syndrome (OHSS), pre-eclampsia and/or premature delivery, which is why proteomic studies have been conducted on PCOS patients. They were able to identify early biomarkers of these complications in an attempt to better diagnose and treat affected women, and thus ensure normal pregnancy.

PCOS patients who wish to have a child often require controlled ovarian stimulation to restore the hormone balance needed to resume ovulation. However, PCOS increases the risk of developing OHSS [16], a potentially life-threatening iatrogenic complication that occurs during assisted human reproduction. For this reason, a more efficient early prediction technique for OHSS is vital. With this in mind, the team of Wu et al. (2017) identified overexpression of haptoglobin, fibrinogen and lipoprotein lipase as early serum biomarkers for predicting OHSS in PCOS patients [16].

A systematic review showed that pregnant women with PCOS were actually four times more likely to develop pre-eclampsia than controls [17]. Therefore, finding the molecular mechanisms underpinning the link between PCOS and pre-eclampsia could serve to minimize the occurrence of maternal and fetal morbidity/mortality associated with this complication. Detecting the relevant biomarkers by proteomic approaches would address this need. Indeed, five biomarkers were found to be differently expressed in women with pre-eclampsia and PCOS compared to controls; transferrin, fibrinogen chain variants α, β and Ɣ, and kininogen-1 were overexpressed, while annexin 2 and peroxiroxin 2 were underexpressed. These biomarkers were found in serum, follicular fluid and ovarian biopsies [17].

Premature births are also a major cause of neonatal mortality and morbidity, and women with PCOS are at a high risk of giving birth prematurely [18]. Research studies are needed to investigate the mechanisms linking PCOS and preterm delivery in order to facilitate detection and develop new preventive strategies. In this context, six biomarkers were found to be similarly overexpressed in women affected by the threat of preterm birth and PCOS compared to controls, including M1/M2 pyruvate kinase (PKM1/M2), vimentin, fructose bisphosphonate aldolase A, HSP27, peroxiredoxin-1 and transferrin [18].

Proteomics offers great potential for acquiring new knowledge to help manage high-risk pregnancies, such as in women with PCOS, but it is not without limits. The relative delay in translating research results to clinical applications is still a cause for concern.

## 4. The Future of Human Ovary Proteomics

### 4.1. In Vivo Mass Spectrometry

Groundbreaking new technology that might provide the impetus to propel MS from the laboratory to the operating room is at an advanced stage of development by Fournier’s team [19]. Their new device, the SpiderMass, consists of a pulsed laser excitation device tuned to heat water in order to create vapor to carry ionized molecules from in vivo tissue to the MS analyzer for measurement and protein identification. The device is already in use in animals (cows, dogs, etc.), but needs final optimization with respect to protein detection rates before clinical implementation.

This technology may finally open the door to noninvasive, real-time in vivo proteomics. Such an advance might enable us to one day build larger ovarian proteomic databases, with references to compare pathological and healthy tissues, and accelerate identification of biomarkers that could potentially change the approach to early diagnosis and tailored infertility treatments. Moreover, it may obviate the need to collect biopsies from volunteer patients in order to run proteomic studies, which will help overcome the difficulties of conducting proteomic analyses on scarce human tissues such as ovarian tissue.

Beyond the potential applications of the SpiderMass device in gynecology, it may someday allow surgeons to pinpoint markers of cancer in a living patient’s tissue during surgery, without having to wait for the pathology report that can take up to 45 risky minutes under anesthesia.

### 4.2. Proteomics to Help Engineer a Transplantable Artificial Ovary

In addition to detecting biomarkers, proteomics is used to monitor the development of cells in contact with synthetic or natural biopolymers both in vivo and in vitro, and to check the suitability of used scaffolds and their safety. In case of bioartificial organs, proteomics can be applied to verify the biomimicry of created organs by comparing their biochemical composition with the real thing. This serves to improve development protocols and ensure their reproducibility before applying them in a clinical context [20]. Indeed, a number of decellularized matrices that have been commercialized and clinically applied have undergone proteomic characterization [21]. A similar approach has been adopted with respect to the ovary. The team of Amorim is currently working on the development of a transplantable artificial ovary, whose main purpose is to safeguard the fertility of leukemia patients who cannot undergo assisted reproduction techniques or grafting of frozen-thawed ovarian tissue due to the risk of reintroducing cancer cells.

The artificial ovary is composed of follicles isolated from cryopreserved tissue and confined inside a physical support, mimicking the natural microenvironment of follicles in the ovary. However, in order to evaluate the quality of this bioengineered environment, a reliable reference is needed. On this basis, any synthetic process used can be monitored and improved, which also allows for the tracking of the composition of the neoformed ECM after transplantation. The team has already succeeded in establishing the most complete proteomic description of the human ovary to date, while at the same time revealing the composition of its ECM [22].

### 4.3. Challenges of Proteomic Studies of the Human Ovary

Available proteomic studies of the ovary rely primarily on the analysis of body fluids such as serum and follicular fluid, or evaluation of ovarian tissue and cells. However, the heterogeneity of biological samples, their collection, storage conditions and preparation make comparison and interpretation of results somewhat problematic.

Although serum is relatively easy to collect given the availability and accessibility of blood samples, the dynamic range of proteins involved—namely the differences in amounts of proteins present and the extreme complexity of the blood proteome—hampers analysis by MS, preventing detection of valuable but scarce proteins in samples [23]. Moreover, a high concentration of albumin and other circulating proteins such as immunoglobulins in serum make separation of proteins difficult. While there are techniques to isolate them by affinity, this approach risks eliminating proteins of interest because of their low specificity to albumin and immunoglobulins [23]. There are also significant variations between serum sample studies, since the composition of blood is likely to alter under the influence of different conditions, such as smoking, age, metabolic diseases, medications and body mass index.

Follicular fluid is likewise easy to collect in a noninvasive way, especially in assisted reproduction centers. However, its limits of application in proteomics do not differ from those of serum, also due to its complexity. The same albumin depletion techniques can be performed, and several studies have actually shown that concentrations of isoforms/albumin fragments in follicular fluid may be able to predict the quality of oocytes and even embryos [7,24,25].

On the other hand, since follicular fluid is usually collected from patients undergoing hormone hyperstimulation, levels of some inflammatory factors could well be affected, which would introduce bias into the interpretation of results. The same is true of signaling pathways with which detected proteins are correlated, such as in tissue repair, in how they complement activation and coagulation [7].

Although analysis of ovarian tissue presents its own challenges (Table 1) mainly related to the invasiveness of the sampling technique and the restriction caused by a small biopsy size, it is more likely to reflect the pathogenesis of ovarian disorders [17] that manifest particularly through periodic changes to the ECM. ECM proteomics can not only reveal novel diagnostic tumor markers and offer new therapeutic leads, it can also enable detection of cellular development effectors in the extracellular space. Improved understanding of the ECM will promote application of reproductive tissue engineering in vivo and in vitro, namely the development of an artificial ovary and creation of biomimetic matrices for maturation of ovarian follicles by 3D culture [26].

However, the question is how can we overcome the complexity of ovarian tissue and manage to extract ECM proteins? Indeed, extracellular proteins present an analytical challenge because of the biochemical properties of their highly crosslinked and therefore insoluble molecules, even in the strongest organic detergents and denaturants. Nevertheless, the high insolubility of the ECM could be exploited to reduce the complexity of tissue samples and detect scarce proteins in the ECM [27]. To this end, it is first necessary to eliminate its intracellular proteins, which are much more soluble, without losing the rest of the proteins in the ECM. This can be achieved by varying salt concentrations and the pH of the tissue homogenate solution.

Once the intracellular proteins have been extracted and removed, we must ensure that total solubilization has been obtained, since only soluble proteins can be analyzed by MS. Incomplete solubilization results in incomplete characterization. Although a number of teams use very strong solvents and detergents to solubilize the matrix, the most common of which is sodium dodecyl sulfate (SDS), these reagents are difficult to remove from samples and are responsible for serious artifacts in the mass spectrometer, even at very low concentrations (<0.01% SDS) [28]. There is therefore a real need to develop appropriate techniques with reproducible and reliable MS-compatible reagents for proteomic characterization of the ECM and biomarker investigation, in order to allow extrapolation of this technology to a clinical setting.

There is an array of potential biomarkers reported in the literature and lists of proteins distinguishing different compartments of healthy and pathological ovaries generated by MS. However, the sheer heterogeneity of analyzed samples and techniques serves to impede validation of these data and their clinical application. Indeed, pre-analytical variables, such as sample processing and specimen storage conditions, should be addressed when comparing different studies, as these factors may have a notable impact on the results [23].

Meticulous selection of pre-analytical sample processing, application of validated techniques (Figure 2) and statistical evaluation of data are essential to generate reliable results. Every effort should be made to ensure that study and control samples are subjected to the same techniques throughout the proteomic process, from sample collection to data analysis [29]. In this regard, clear guidelines are needed to design clinically applicable proteomic analyses.

Appropriate bioinformatic analysis is key to accurate interpretation of results, since management, evaluation and interpretation of large volume of raw and processed data require a combination of different bioinformatic techniques, from data mining to comparison of data sequences and predictive models. Data acquisition, filtering and normalization are also important, and each of these steps can have a profound effect on the results. Complementary validation techniques, such as immunoassays or western blots, need to be employed before formulating physiopathological hypotheses based on quantitative (Figure 2) or qualitative differences in proteins [30].

### 4.4. Proteomics Is Not an Island: The Added Value of Proteogenomics

Proteins serve as a critical link between genotypes and phenotypes. Proteomic profiles reflect cellular responses to genomic, epigenomic and environmental alterations, and in turn shape these responses. Thus, proteomics is not an island entirely of itself. Proteogenomics has now emerged as a rapidly evolving field at the intersection of genomics, transcriptomics, and proteomics. Its essential goal is integration of multiomic data for accurate annotation and reciprocal refinement of genomic and proteomic models [31]. This integrative approach has the potential to provide solid evidence for translation of previously unknown transcripts. Indeed, newly reported peptides can represent single amino acid variants, splice variants, gene fusions, RNA editing events, novel open reading frames, translated noncoding RNAs and pseudogenes, among many other things. Proteogenomic platforms can be used to investigate which of these novel events gets translated at the protein level, making them potential candidates for new druggable targets or new diagnostic or prognostic biomarkers for a wide spectrum of diseases, including ovarian cancer [32]. Indeed, this approach is increasingly used in oncology, since many patients do not respond as predicted to therapies recommended based on genomic profiles of their tumors [31]. Recent studies pairing genomic and proteomic analyses in the same tumors have shown that the proteome encodes novel information that cannot be discerned by genomic analysis alone. Moreover, from a clinical perspective, the majority of molecular therapies do not target the patient genome, but rather cellular and extracellular proteins (kinase inhibitors, PARP, immunomodulatory proteins, etc). The National Cancer Institute (NCI) has funded a major initiative in cancer proteogenomics through its Clinical Proteomic Tumor Analysis Consortium (CPTAC). The CPTAC flagship reports on ovarian cancer have provided a resource for the cancer research community, both by indicating which of the genomic and transcriptomic features associated with this cancer was recapitulated at the protein level, and by providing new insights into the substantial impact of post-translational modifications, specifically phosphorylation and acetylation, on the functional activities associated with DNA repair, proliferation and survival. Furthermore, proteogenomics can complement proteomic databases used for protein sequencing and identification (such as rEFsEQ or Ensembl), which are unable to identify any novel cancer-specific sequence [33]. This can be achieved by incorporating candidate protein sequences derived from exome sequencing or RNA-Seq data into a customized database for MS/MS spectra interpretation, allowing identification of sample-specific peptides that are missing from the reference database, or even protein isoforms. Using this approach, the CPTAC studies into ovarian cancer have provided proteomic validation for 14 somatic mutations [34]. In a few cases, it has been shown that protein profiling is more closely aligned with function than mRNA profiling data, suggesting that protein translation and degradation are tightly regulated and play a critical role in determining gene function [35].

On the other hand, for many genes, mRNA measurements are considered poor predictors and very few mRNA biomarkers have been translated into clinical practice, however, new approaches are emerging. The use of somatic copy-number alteration, DNA methylation, mRNA, microRNA and reverse phase protein array (RPPA) may well fulfill the promise of protein measurement in a clinical setting by providing novel insights into multilevel gene expression regulation, signaling networks, disease subtypes and clinical predictions [36]. However, to bridge the gap between proteogenomic biomarker candidate detection and clinical validation, greater precision and highly sensitive techniques are required, especially on a one-cell scale. Until recently, traditional sequencing methods were only able to determine the average of large number of cells, but were incapable of analyzing small numbers or acquiring cellular heterogeneity information.

Single-cell sequencing technologies can sequence and quantify the whole genome of germ cells and embryonic cells at the single-cell level. This will help elucidate the emergence of germ cells and facilitate screening, diagnosis and treatment of reproductive and genetic diseases [37]. Detection of female egg cells, polar cells or embryonic cells by single-cell sequencing to select healthy embryos for transfer can reduce the number of newborns with congenital genetic diseases and help prevent genetic diseases. Li et al. applied genome-wide mapping of human embryos prior to implantation by single-cell multi-sequence sequencing [38]. This study has implications for the analysis of complex and highly-coordinated epigenetic processes involved in the pre-implantation development of human embryos. Vento-Tormo et al. performed a transcriptome analysis of placental cells in early pregnancy by single-cell sequencing technology and mapped placental cell maps. Through the cell map, three subpopulations of perivascular and stromal cells located in different decidual layers and dNK (decidual natural killer) were found [39]. It also identified regulatory factors that may minimize harmful immune responses to the placenta, and interactions contributing to placental competence and reproductive success. These findings are important for understanding early pregnancy processes and improving the diagnosis and treatment of pregnancy-related disorders. The first step towards clinical translation of these genomic discoveries is validation of their expression at the protein level. Although single-cell genome sequencing preceded single-cell proteomics, the latter has so far achieved significant milestones, highlighted in the review by Couvillion et al. [40].

Clearly, with the help of multi-omics, we can shed further light on physiological processes and pathological mechanisms by deciphering how information flows from gene to protein. However, the field of female fertility is not yet taking full advantage of these applications. Moreover, multi-omics approaches still have issues, such as cumbersome utilization and high detection costs, which limit the promotion of this technology. The hope is that techniques will become increasingly simplified and effective and detection costs will fall, and thus these tools can be applied to basic research and play a key role in clinical diagnosis and fertility assessment (Figure 3).

## 5. Conclusions

Proteomics has emerged as a powerful tool, with applications ranging from biomarker identification for effective embryo and oocyte selection and high-risk pregnancy management, to tissue engineering guidance. However, while a number of data sets have been generated in basic research, such data cannot be used clinically. It is therefore essential to translate these new findings into clinical applications, whose impact could significantly enhance patient health and care.

## Figures and Tables

**Figure 1 ijms-20-04209-f001:**
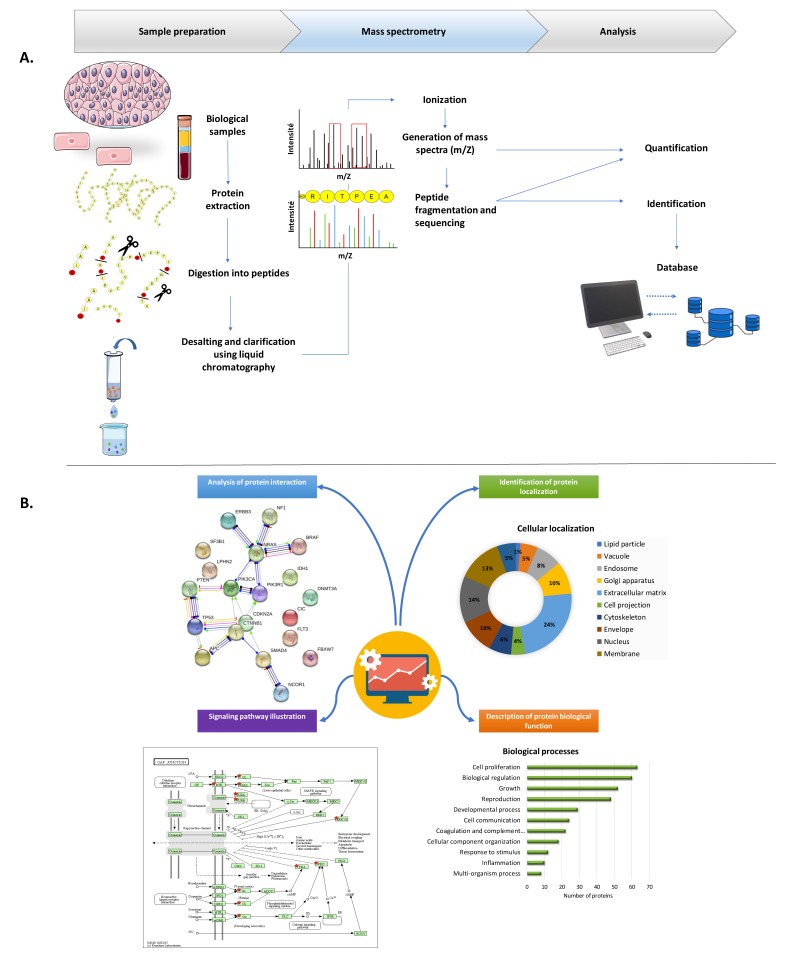
Sample preparation and proteomic data analysis at a glance. (**A**) Standard protocol for proteomic analysis. Proteins extracted from biological samples (tissue, cells, serum, etc.) are first digested by a proteolytic enzyme (generally trypsin). The peptides generated are clarified and separated by liquid chromatography according to their hydrophobicity and/or hydrophilicity. The eluted peptides are then ionized as soon as they enter the analysis unit of the mass spectrometer. This step will further fractionate the peptides and give them mobility that will depend on their charge (z) and molecular weight (m). These two parameters are detected and acquired by mass spectrometry (MS). The peptides are subsequently fragmented and sequenced to obtain information on their amino acid chain. The resulting spectral data allow identification of the starting proteins through use of databases. (**B**) Proteomic data processing through bioinformatic analysis. Bioinformatic tools available today make it possible to handle hundreds or even thousands of proteins detected by MS. Applications may include determining the localization of identified proteins and characterizing biological processes, signaling pathways in which they are involved and possible interactions that bind the different proteins, as well as streamlining potential new physiological mechanisms. More information on open source bioinformatics tools can be found in [2].

**Figure 2 ijms-20-04209-f002:**
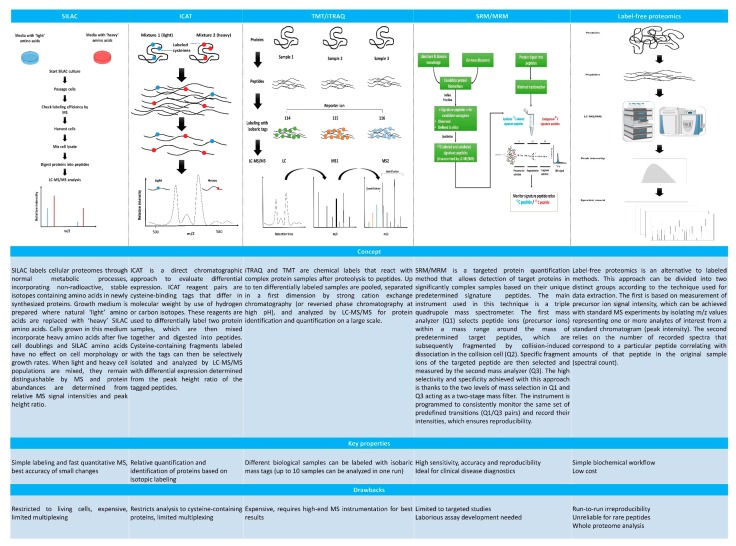
Key features of the main quantitative MS methods. Full names of abbreviated techniques: stable isotope labeling of amino acids (SILAC); isotope-coded affinity tags (ICAT); tandem mass tag (TMT); multiplexed isobaric tags for relative and absolute quantification (iTRAQ); selected reaction monitoring (SRM); multiple reaction monitoring (MRM) and label-free quantification (LFQ).

**Figure 3 ijms-20-04209-f003:**
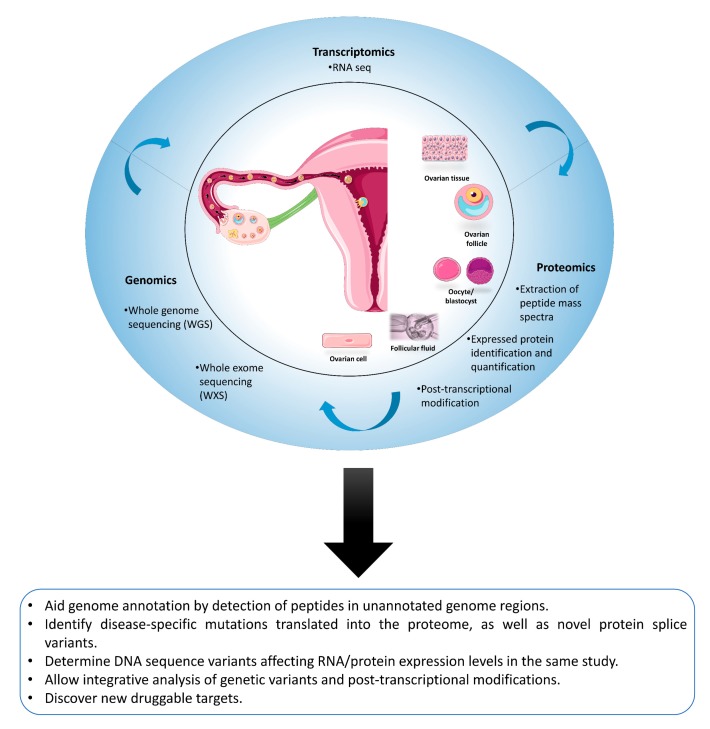
Multi-omics perspectives. When combined, the coding proteome and noncoding transcriptome represent products of the sequence-to-phenotype continuum (DNA to RNA to protein). By combining genomic, transcriptomic and proteomic technologies in the same workflow, these technologies can inform each other and complement our knowledge of the human ovary and procreation.

**Table 1 ijms-20-04209-t001:** Challenges of biological sample proteomics in gynecology.

Difficulty detecting scarce proteins in complex biological samples. Introduction of bias into the interpretation of results due to inappropriate sample preparation protocols. Heterogeneity of proteins detected because of different experimental protocols and proteomic platforms used. Need for corroboration of proteomic results by other techniques before validating a hypothesis. Available protocols not suitable for analysis of small quantities of biological material, especially needed when analyzing scarce samples such as ovarian tissue biopsies.

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
