# Peer review of "The Human Ovary and Future of Fertility Assessment in the Post-Genome Era"

_ijms, 2019, doi:10.3390/ijms20174209_

Round 1

Reviewer 1 Report

Dear Authhors, i really appreciate your manuscript regarding an important theme as the proteomics in the field of the reproductive medicine.

I believe that the work is original, mainly due to the applicability of proteomics in pre-implant diagnostics during an IVF treatment. The possibility to understand the protein expression profile of the follicular fluid for the embryo competence and the oocyte development and maturation could be an important aid in the managment of the infertile women. For these reason the article is acceptable for publication.

It is well written and organized.

Author Response

We thank the reviewer for his/her positive comments. Although the manuscript was already revised by a native English speaker before the first submission, it has now undergone further revision, as requested by the reviewer.

Reviewer 2 Report

The author needs to provide more information It would be great if they could provide information regarding some recent advances in the field such as microRNA, long noncoding RNA etc. Needs to provide more figures and tables Please try to provide latest references in the field Please provide references to the figure panels. For example the network plot shown in Figure 1B please provide the citation

Author Response

“The author needs to provide more information It would be great if they could provide information regarding some recent advances in the field such as microRNA, long noncoding RNA etc.”

We thank the reviewer for his/her suggestions. However, our review focuses primarily on proteomics, which is an underinvestigated field in reproductive biology compared to transcriptomics. RNA is therefore beyond the scope of our manuscript. We have nevertheless integrated a new section describing the perspectives of proteogenomics: the synergetic relationship between genomics, transcriptomics, and proteomics.

The added section is entitled “4.4 Proteomics in not an island: the added value of proteogenomics” (L334-407).

“Needs to provide more figures and tables Please try to provide latest references in the field Please provide references to the figure panels. For example the network plot shown in Figure 1B please provide the citation”

In response to the reviewer’s suggestion, a second table and figure have been added.

Table 2 summarizes key features of the main quantitative MS methods currently used.

Figure 2 highlights the benefits of a multi-omic approach and the complementarity of genomics, transcriptomics and proteomics.

All the figures were designed by the authors and not derived from other manuscripts, which is why no references were provided. However, we have cited a review on open source bioinformatic tools, in the legend to Figure 1.B, as shown below:

Figure 1. Sample preparation and proteomic data analysis at a glance. (A) Standard protocol for proteomic analysis. Proteins extracted from biological samples (tissue, cells, serum, etc) are first digested by a proteolytic enzyme (generally trypsin). The peptides generated are clarified and separated by liquid chromatography according to their hydrophobicity and/or hydrophilicity. The eluted peptides are then ionized as soon as they enter the analysis unit of the mass spectrometer. This step will further fractionate the peptides and give them mobility that will depend on their charge (Z) and molecular weight (m). These two parameters are detected and acquired by MS. The peptides are subsequently fragmented and sequenced to obtain information on their amino acid chain. The resulting spectral data allow identification of the starting proteins through use of databases. (B) Proteomic data processing through bioinformatic analysis. Bioinformatic tools available today make it possible to handle hundreds or even thousands of proteins detected by MS. Applications may include determining the localization of identified proteins and characterizing biological processes, signaling pathways in which they are involved, and possible interactions that bind the different proteins, as well as streamlining potential new physiological mechanisms. More information on open source bioinformatic tools can be found elsewhere (2).

Reviewer 3 Report

Comments to the Author
In manuscript, entitled “ The human ovary and future of fertility assessment in the post-genome era” by Ouni et al., authors want to present the scope of proteomic applications through an overview of the technique and its applications in assisted procreation. This paper is very well written. Moreover, this appears that such important issue as oocyte and embryo quality analysis for their selection for IVF technique, is also well documented and well described. The data are of interest and important for research teams have focused their efforts on assisted procreation techniques. In my opinion, the article is printable.

Author Response

We thank the reviewer for his/her positive and appreciative comments.

Round 2

Reviewer 2 Report

The authors have addressed all the comments.